# The Efficiency of National Innovation Policy Programs: The Case of Serbia

Sonja Đuričin *, Isidora Beraha, Olivera Jovanović, Marija Mosurović Ružičić, Marija Lazarević-Moravčević and Mihailo Paunović

Innovation Economics Department, Institute of Economic Sciences, 11000 Belgrade, Serbia;
isidora.beraha@ien.bg.ac.rs (I.B.); olivera.jovanovic@ien.bg.ac.rs (O.J.); marija.mosurovic@ien.bg.ac.rs (M.M.R.);
marija.lazarevic@ien.bg.ac.rs (M.L.-M.); mihailo.paunovic@ien.bg.ac.rs (M.P.)
* Correspondence: sonja.djuricin@ien.bg.ac.rs

**Abstract:** We aimed to assess the efficiency of the selected national innovation policy programs in the Republic of Serbia. We analyzed the impact of the Innovation Fund's Mini-Grants and Matching Grants programs on the operating revenue growth of beneficiary micro, small, and medium enterprises. An econometric analysis of panel data was conducted. Because of the small number of periods observed, a model of individual effects was applied. Conclusions and recommendations were based on the results of random effects models. The findings indicate that program funding increased business revenues compared to the period before and that there was a direct link between indebtedness and revenue growth, which confirmed the positive impact of financing on the sustainable development prospects of beneficiaries through facilitating access to funding and innovation capacity improvement. These findings can have important policy implications as they provide guidelines for designing future actions and empirically confirm the need to increase public expenditures for innovation policy.

**Keywords:** national innovation policy programs; efficiency; sustainable development; micro; small and medium enterprises; financing; operating revenue; indebtedness; the Republic of Serbia

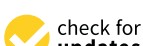



## 1. Introduction

The policy argument for government funding of innovation activity is tied to addressing the problem of a lack of funds as one of the most limiting factors affecting an enterprise's ability to innovate [1]. In today's highly competitive environment, innovation is crucial for the overall success of an enterprise, paving the way for long-term competitive advantage [2–5]. Because an economy's ability to generate innovative and high-tech items indicates its long-term competitiveness and development potential, innovative entrepreneurial enterprises have long been recognized as one of the key driving forces of long-term economic growth [6]. SMEs have the potential to be key sources of innovation, structural change, and industrial renewal; nevertheless, they are often hampered by several factors, the most prominent of which is funding. Innovation policy instruments must be designed and mixed in such a way that they address the innovation system's problems [7]. The authors in [8] emphasized that the ranking and classification of a country in the European Innovation Scoreboard should be reflected in policies aimed at addressing the country's various innovation challenges, structural characteristics, and policy needs. However, despite these assumptions, the authors in [9] revealed a surprising consistency in European countries' innovation policy mixes. The evidence needed to guide policy decisions is often lacking. Methods and data for capturing and analyzing different forms of innovations are limited, and governments often lack the capacity to monitor their own policy instruments [10]. This highlights the need for more research into which support mechanisms are most efficient in helping innovative SMEs overcome their challenges. The purpose of this paper is to discuss the arguments for public funding of innovation activities, to draw the attention of

academics and policy makers to the importance of gathering evidence to design policies that solve actual problems in the innovation system, and to add to the limited body of knowledge on how to evaluate the efficiency of existing innovation policy programs. This study provides evidence that will assist in the more efficient use of limited budget resources to foster innovation in Serbia, as well as in other countries that can apply the proposed methodology to evaluate the efficiency of their programs.

The focus of this research is on a quantitative assessment of the effects of innovation policy programs on the operating revenue growth of SMEs, which gives it added value because the number of studies that examine the relationships between specific national innovation support programs and the business performance of beneficiaries is still insufficient [11]. Even though post-communist countries' access to finance for enterprises remains limited, empirical evidence of the impact of policy programs in Central and Eastern Europe is modest [12].

The objective of the research is to assess the efficiency of the two programs of the Innovation Fund of the Republic of Serbia, i.e., the Mini-Grants Program and the Matching Grants. Specifically, we quantitatively analyze the link between public funding through two innovation policy programs and the operating revenue growth of beneficiary micro, small, and medium enterprises. The research subjects are the beneficiaries of the two programs of the Innovation Fund.

The research context is determined by Serbia's innovation system, as a framework containing legislation, interactions, and relationships among diverse actors; innovation policy measures; and all other elements that directly or indirectly influence the circumstances in which an innovation is realized [13]. Even though certain progress was made in the innovation segment, it should be outlined that "the investment in research and development in the Republic of Serbia with a share of 0.89% of GDP in 2019 was more than twice lower than the EU average (2.19% GDP) and the standard provided by the Lisbon Convention (3% of GDP)" [14]. On the other hand, the EU spends 0.8% of GDP less on research and development each year compared to the USA and 1.5% less than Japan [15]. According to a study of 3709 SMEs conducted between 2018 and 2020, more than 69% of large organizations, 58% of medium-sized enterprises, and 54% of small businesses are engaged in innovation activities, and the main barriers to engaging in innovation activities are high expenses and a lack of financial resources [16]. According to the Law on Innovation Activity, innovation policy in Serbia entails setting goals and creating systemic conditions for the conception, development, and implementation of innovations. The program of innovation activities is used to implement and achieve the innovation policy through innovation and development initiatives. Only entities registered in the Register of Innovation Activity are eligible to receive state incentive measures and budget funds for the development of innovation activity. As the only state agency committed to helping innovative activities and administering financial resources to stimulate innovation, the Serbian Innovation Fund plays an important role in accomplishing the research and innovation policy. It has granted EUR 51.2 million for 317 innovative projects over its 11-year lifetime. Accordingly, the role of the fund is significant in achieving the research and innovation policy. Given Serbia's low level of national investment in research and innovation, enhancing its understanding of the efficiency of the innovation policy is necessary.

The research begins with the following hypothesis:

**Hypothesis 1 (H1):** *Funds allocated through the Mini-Grants Program and Matching Grants Program affect the operating revenue growth of beneficiary micro, small, and medium enterprises.*

Accordingly, we conducted an econometric panel data analysis. A model of individual effects was used due to the limited number of observed periods. The results of the random effects models serve as the foundation for conclusions and recommendations.

The efficiency of using funds from the national innovation programs is observed through the growth of operating revenues as one of the primary factors of profitability. An

increase in beneficiaries' operating revenue confirms the selected programs' efficiency. Operating revenue in absolute amount is used as an indicator of the profitability of beneficiary enterprises, as explained in more detail in Section 3. As control variables, the analysis uses total assets and indebtedness. The growth in operating revenue is a prerequisite for increasing the ability to provide the necessary funding and consequently improve innovation activities and achieve sustainable development [17].

The paper contributes to the literature in two ways. First, we applied a methodology that examines the direct relationship between public funding and an increase in beneficiary enterprises' operating revenues. It is a rare attempt to quantitatively evaluate the impact of innovation policy programs. Furthermore, the research findings provide solid insight for policy makers to act in long-term directions by allocating more funds for initiatives that positively impact the business performance of SMEs.

The remainder of the paper is structured as follows. Section 2 presents the theoretical background, including the relevance of the literature gap we intend to address; Section 3 presents the methodology and variables used to assess the efficiency of the selected programs of the Serbian Innovation Fund in the period 2011–2020; Section 4 summarizes the findings of the research question that guided this research; Section 5 discusses the main findings supported by existing research results; Section 6 concludes with the policy implications in Serbia and other countries with similar levels of innovation system development. The discussion section of the paper explains the link between indebtedness, which is represented by the total liabilities of enterprises, and operating income growth.

## 2. Theoretical Background

Innovation policy aims to create a conducive environment for bringing ideas to the market by bridging the gap between research and technological development policy and industrial policy [18]. National innovation policy is gradually shifting the focus from obtaining an institutional context for performing innovation activities toward meeting sustainable development goals [19,20]. The issue of the allocation of public funds to innovation policy areas must be related to assessing the efficiency of existing programs and measures from the perspective of achieving sustainable development. SMEs' sustainable development is perceived as achieving enterprise development [21]. Despite this relatively narrow understanding of the sustainable development concept, it focuses on innovation as a critical factor for enterprise development. In the context of innovative SMEs, it is a question of how much public funding contributes to their development.

Having SMEs at the core of national innovation policies is necessary for sustainable development. This, however, is not always the case for many reasons. For example, according to [22], China's national innovation system pays little consideration to the long-term viability of SMEs' innovative operations for two reasons. Firstly, the national innovation system's scope is narrowly defined. Secondly, the top-down, government-oriented R&D system, which focuses on large state-owned businesses, offers little room for SMEs to pursue innovation policies.

It is empirically proven that funding opportunities determine the innovation activities of SMEs [23]. The lack of financial resources is usually the most limiting obstacle for SMEs [24]. Firstly, innovative enterprises have many intangible assets that cannot be pledged as collateral [25]. Limited access to external finance can also be attributed to the low profitability of financial success for their R&D projects, resulting in uncertainty about their return [26]. Moreover, the problem of information asymmetry is one of the most significant barriers to obtaining external finance. Information asymmetry can lead to financial constraints, especially for innovative enterprises with limited internal financial resources, such as small and young businesses [27].

Bank loans for business expansion are frequently difficult for SMEs to obtain, especially in developing countries. This highlights the need to provide SMEs with a variety of financing options. The European Commission launched the SME Instrument in 2014 to assist enterprises with strong growth potential that require external financing. SMEs that

participated in EU-funded projects produced a significant number of innovations and saw increases in their turnover and employment [28].

The EU budget is spent in areas where financial support has the greatest impact [29]. The European Commission provides direct funding in the form of reimbursable or non-reimbursable aid to organizations and projects that promote European interests and are involved in implementing EU policies and programs [30,31]. The evolution of priorities within the EUs' innovation policies points to various priorities that have changed over time, starting with measures supporting primarily science industry links at the beginning of the 2000s and moving on to high-technology sectors, commercializing university research, technology parks, etc., since 2006 [32]. Research reveals that most member states appear to use similar combinations of innovation policy instruments, regardless of their innovative position and the difficulties they encounter (competitive public research funding, collaborative RDI programs, direct business support for R&D, direct support for innovation, loans for firms, and tax incentives) [33]. Evaluations of innovation policy instruments are often focused on implementation concerns and evaluate target achievement, with the focus being on how well projects and activities align with the programs' goals and how effectively and efficiently they are carried out [34]. Typically, evaluation systems quantify innovation in terms of inputs (e.g., "promising practices" or new technologies) and then estimate the likely value added associated with such interventions in terms of firm-level (such as patents or interfirm collaborations) and regional measures (such as jobs created and safeguarded) [35–38]. However, it has been challenging for all member states to measure the impact of EU-funded programs [39]. The authors in [40] demonstrated that a public program funded from the EU is important for innovation among SMEs, that direct support for R&D across the EU countries has some positive impacts on labor productivity for innovative firms [41], that specific EU country policies focused on corporate R&D have had positive spillover effects [42–44], and that public funding has benefits for private R&D [45]. The authors in [46] underlined that using EU structural funds enhances the potential of SMEs to obtain a better competitive position in the market.

In assessing the impact of structural funding on SMEs' competitiveness, evidence shows differences amongst beneficiary regions across the EU. To improve efficiency, national programs should reduce their reliance on EU financing rules, while regional programs should have more efficient implementation [47].

The research presented in this paper assists in evaluating Serbia's innovation policy measures and supports the development of funding programs tailored to SMEs' needs. Scientific knowledge about the efficacy of public grants to innovative enterprises is limited [48]. The scientific literature has been primarily incapable of providing policy makers with clear instructions for identifying the best mechanisms for each specific institutional context [49,50]. Surprisingly little attention has been paid to the evaluation processes through which public funds are allocated and, in particular, to the potential for differing allocation mechanisms to generate backlash in subsidy effectiveness [51]. Moreover, subsidies have a positive and significant treatment effect on new technology-based enterprises' total factor productivity growth, but only if subsidies are provided competitively and are targeted to increase R&D investments. The authors in [52] used a proprietary sample of 129 startups located in eight incubators to find that securing an early grant enhances the rate at which initiatives gain private investment capital but not revenue over time. The authors in [53] revealed that subsidies are an important policy instrument in encouraging young innovative companies to develop inventions. The results of a survey on SMEs in Belgium conducted by the authors in [54] and the results of the following studies [55] suggest that national support programs increase the beneficiary's possibility of both equity financing and access to commercial bank loans.

The authors in [56] assessed three potential types of effects of Germany's most comprehensive place-based innovation policy, the Innovative Regional Growth Cores, on a wide range of enterprise and regional outcomes: (1) the policy's effects on directly subsidized firms; (2) spillover effects on non-subsidized innovative firms in the same region;

and (3) (aggregate) effects on regional-level economic outcomes. They found that directly treated firms increase their R&D activities in the medium term, whereas using a variety of econometric methodologies failed to produce significant or economically meaningful evidence for the effectiveness of channels (2) and (3). The authors in [57] found that public funds for innovation have short-term effects on a firm's capabilities, medium-term effects on innovation efforts, and long-term effects on productivity in their study on the impact of public funds for innovation on a firm's capabilities, innovative dynamics, and economic performance.

There have been studies into whether receiving an innovation or R&D grant has a positive certification effect that facilitates SMEs' subsequent access to debt and equity financing [58], but what distinguishes this research is the attempt to quantify revenue growth as a result of increased corporate indebtedness.

The research presented in this paper offers policy makers useful information for enhancing beneficiaries' business performance. Policy makers may be assisted in promoting effective policy governance and the implementation of policies by evaluating the efficiency of the national innovation policy programs. Additionally, this is important for developing countries where the practice of innovation policy is still evolving and institutional reform is required [59].

## 3. Materials and Methods

### 3.1. Materials

Assessing the efficiency of using funds from the Mini-Grants Program and Matching Grants Program requires the generation of appropriate data with reasonable reliability. Data were retrieved from the Bisnode database and the website of the Innovation Fund of the Republic of Serbia. The Bisnode database generates data from annual financial reports and other documentation submitted by enterprises in the Republic of Serbia to the Business Registers Agency. Data from the Bisnode database are not publicly available and were obtained with appropriate financial compensation. On the other hand, data on the amount of funds allocated to beneficiaries of the Mini-Grants Program and Matching Grants Program are publicly available on the website of the Innovation Fund of the Republic of Serbia [60].

The use of funds from the Mini-Grants Program and Matching Grants Program is possible if several requirements are fulfilled. The first requirement, which is applicable to both programs, evaluates the possibility that the project will be able to generate revenue from innovation 2 to 3 years after it starts. The second requirement applies to the Mini-Grants Program and implies that the enterprise was founded a maximum of 3 years ago. This means that funds are approved for young enterprises in the initial stages of development. Furthermore, only micro and small enterprises are eligible to participate in this program. As for the Matching Grants Program, the beneficiaries can be older enterprises, i.e., operating for more than 3 years, and they can be micro, small, or medium-sized.

All individual data from the available database were considered in detail to eliminate extreme values, outliers, and other data that could impair the reliability of the conclusions. Of the initial 39 enterprises, 36 were included in the final analysis. Due to the verification of individual financial variables, the existence of three outliers was determined. For this reason, the final sample consisted of 17 enterprises participating in the Mini-Grants Program and 19 enterprises participating in the Matching Grants Program. The analysis covers the period 2011–2020. The study includes the business year preceding the disbursement of funds, the business year in which the funds were granted, and 3 years after the distribution of funds. The beginning and end of the observation period are determined by the year in which the funds were approved for an enterprise. The time analysis entails 5 years; hence, we have unbalanced panel data.

The funds from the programs were approved in cycles. In the observed period, each program was realized through four cycles. According to publicly available data on the website of the Innovation Fund, the lowest amount of funds under the Mini-Grants

Program was approved in the first cycle (EUR 224,486). In contrast, the highest amount was approved in the fourth cycle (EUR 600,230). Under the Matching Grants Program, most funds were approved in the first cycle (EUR 1,285,043), while the least funds were approved in the third cycle (EUR 552,485).

### 3.2. Variables

Determinants of enterprise profitability were divided into internal and external [6,61,62]. The research objectives and the available data observed only the internal determinants of profitability. Although relative profitability indicators such as return on assets (ROA) and return on assets equity (ROE) are much more commonly used in the literature [6,63–66], when there is a lack of available data of a reasonable level of reliability, it is possible to use absolute indicators of profitability, as in [67]. Ratio indicators help compare the profitability of enterprises operating in the same sector, which was not the case in this study. In addition, as the sample consisted of young enterprises, their financial statements often lacked data to calculate relative profitability measures, or these data were unreliable. For example, a significant number of beneficiaries in the sample had a negligibly small, almost zero, amount of equity that implied the extreme values of their ROE. Moreover, although young, newly established enterprises are not expected to generate a significant value of financial income and expenses, the research in some cases found a substantial difference between the amount of operating profit and net profit. For these reasons, operating revenue was selected as the dependent variable. Operating revenue is the result of the enterprise's core business and was chosen by the research objective and requirements of the Mini-Grants Program and Matching Grants Program.

In addition to the dependent variable, one independent and two control variables were used in the study. The primary independent variable was represented by the funds approved under the Mini-Grants Program and Matching Grants Program. In an econometric analysis, it is expressed by a *dummy variable*, which takes the value 1 if the beneficiary received funds in that year; otherwise, it takes the value 0. The first control variable was an internal determinant of profitability, which approximates the size of the enterprise and refers to total business assets [6,63,64,68]. The second control variable was corporate indebtedness. Although financial leverage is most often used in the literature to estimate indebtedness, absolute amounts were used in this research due to the previously mentioned limitations of relative indicators. Indebtedness is represented by the enterprise's total liabilities [67–69].

### 3.3. Methods

Panel analysis was used for the data analysis. Defining an appropriate analysis model is usually based on the number of observation units and the period's length. In this research, the number of years was 5 for each enterprise, and the number of observation units was 36. As the number of observation units was greater than the number of years, econometric analysis of classical panel data was applied. The database consisted of data where the dependent and selected control variables varied in two dimensions: temporal (t) and individual (s). The values for observation units were recorded in five consecutive periods. Observation units were beneficiaries of the Mini-Grants Program and Matching Grants Program. The time coverage was not the same for all observation units but depended on when the enterprise became a beneficiary of the program.

The evaluation of the effects of funds from the Mini-Grants Program and Matching Grants Program on the financial result of the beneficiary started with a model of individual effects, which is justified when there is a small number of periods [70]. This model evaluates the effects of individual variables not explicitly included in the model on the dependent variable variations by observation units. Two models are distinguished in the literature [70]: the fixed effects model (FE model) and the random effects model (RE model). According to [71], for the use of the model of fixed effects, the independent variable must vary between individuals and in time. In contrast, for the application of the model of random effects, it is

necessary to fulfill the assumption of no correlation between random effects and regressors. If both premises are met, the estimates of both models are unbiased and consistent, with the estimates of random effects models being more efficient because they have lower variance [70].

The application of an appropriate model is based on the assumptions made by [72], according to [73], indicating that the model of fixed effects is more adequate when the number of years observed is small, when the number of observation units is large, and if observation units are not randomly selected in the sample.

The first step in the empirical analysis of this research was the estimation of the simple regression model. Since the estimation of the effects of approved funds on operating revenue was the main objective of the research, the variable of funds was considered the most important. In the second and third steps, the regression models included control variables using availability and literature review criteria. Quantitative analysis in this paper was limited by the number of observations, where more than two independent variables could not be included in the model. In the second step, variable *lnassets*, which approximates the size of the enterprise, was included as a control variable. In the third model, variable *lnliabilities,* which approximates the indebtedness of the enterprises, was included as a control variable. The three models were expressed as follows:

$$lnrevenues_{it} = \alpha_i + \beta_{it}funds + u_{it} \tag{1}$$

$$lnrevenues_{it} = \alpha_i + \beta_{it}funds + \beta_{it}lnassets_{it} + u_{it} \tag{2}$$

$$lnrevenues_{it} = \alpha_i + \beta_{it}funds + \beta_{it}lnliabilities_{it} + u_{it} \tag{3}$$

where $lnrevenues_{it}$ is the logarithmic of value of the operating revenue per observation unit in the period $t$, $\alpha_i$ is a constant, $\beta_{it}$ is an unknown regression parameter, *funds* is a dummy variable for funds received from the programs (1 if the enterprise received funds, 0 otherwise), $lnassets_{it}$ is the logarithmic value of the total business assets per observation unit in the period $t$, $lnliabilities_{it}$ is the logarithmic value of the corporate indebtedness per observation unit in the period $t$, and $u_{it}$ is a composite random error ($i = 1, \dots, 36, t = 2011, \dots, 2020$).

The effects of the approved funds on the financial performance of enterprises were assessed in the entire sample, examining the beneficiaries of both programs together. Due to the small number of observations per beneficiary group, it was impossible to determine the effects of programs individually.

Since we examined a short time series that consisted of only five time periods, it was assumed that the stationarity issue did not arise. Therefore, the stationarity test for all variables in the model was not used. The Hausman test was used to determine the nature of the individual effects, i.e., fixed or random effects. The Jochmans test (portmanteau test for correlation in short panels) was used to test the serial correlation, while the Breusch and Pagan Lagrangian multiplier test was used to test heteroskedasticity. Since the sample size was larger than 30, the error term was approximately normally distributed according to the central limit theorem [74].

According to [75], the use of panel analysis has multiple advantages in research, such as control of individual heterogeneities, lower occurrence of multicollinearity between variables, a greater degree of freedom, greater efficiency of assessments, enabling monitoring of phenomena, and determining characteristics of their behavior over time and according to observation units.

## 4. Results

Hausman test values are given in Table 1. The Hausman test determined the nature of the individual effects. According to the calculated value of $\chi^2$ statistics for each model and the corresponding *p*-value, we concluded that econometric analysis could be continued using a model with random effects. The results for all three models are shown in Table 2.

**Table 1.** Hausman's choice of estimators of individual effects.

| Type of Model | $\chi^2$ Statistics | *p*-Value | Source |
|:---:|:---:|:---:|:---:|
| Model 1 | 1.45 | 0.2289 | Random effects model |
| Model 2 | 0.65 | 0.7217 | Random effects model |
| Model 3 | 4.99 | 0.0824 | Random effects model |

Source: Authors' calculation.

**Table 2.** The results of the analysis.

| | **Model 1** | **Model 2** | **Model 3** |
|:---:|:---:|:---:|:---:|
| Funds | 0.21195 (0.23792) | 0.24164 (0.16939) | 0.25439 * (0.02617) |
| *lnassets* | | 0.80115 * (0.04573) | |
| *lnliabilities* | | | 0.66681 * (0.05426) |
| Const | 9.82798 (0.32355) | 2.18996 (0.45694) | 3.79974 (0.53216) |

* Statistically significant results ($p < 0.05$). Source: Authors' calculation.

The results of the estimation of regression parameters for models 1, 2, and 3 are given in Table 2. The estimated parameter of funds in the first regression model was positive, indicating that the use of program funds increases operating revenues. However, the corresponding *p*-value (0.373) was greater than 0.05; hence, this variable was not statistically significant. The value of Wald's test statistics ($\chi^2$ (1) = 0.79, $p = 0.3730$) indicated that additional independent variables should extend this model. Since the null hypothesis of the Wald test was rejected, we concluded that this model specification was inappropriate. The analysis should include control variables, performed through models 2 and 3.

In model 2, a control variable *lnassets* was introduced, which approximates the size of the enterprise. Although the estimated parameter of *lnassets* was positive and statistically significant ($p < 0.05$), its introduction did not significantly affect a change in the results or statistical significance of the funds variable. In other words, the value of the estimated parameter of funds remained positive but did not become statistically significant. The conclusion from model 2 is that the enterprise's size has a positive and statistically significant impact on operating revenue. The growth of business assets by 10% on average led to an increase in operating revenue by about 8%. Wald's test statistics for model 2 ($\chi^2$ (1) = 308.11, $p = 0.000$) indicated good model specification.

A control variable *lnliabilities* was introduced in model 3, approximating the indebtedness of enterprises. Its introduction significantly affected the results since the estimated parameter of funds remained positive, but became statistically significant. In other words, the funds received from the programs positively affect the growth of the operating revenues of beneficiaries. In addition, the liabilities variable coefficient was positive and statistically significant, indicating that an increase in corporate indebtedness by 10% on average led to an increase in the enterprise's operating revenue by 6.7%. Wald's test statistics for model 3 ($\chi^2$ (1) = 51.37, $p = 0.000$) indicated good model specification.

The serial correlation was tested using the Jochmans [76] test (portmanteau test for correlation in short panels) [76,77]. The results showed no serial correlation ($p > 0.05$) in any model. Heteroskedasticity testing was performed using the Breusch and Pagan Lagrangian multiplier test [78]. Since the test results were statistically significant ($p < 0.05$), the variance of random errors was not constant, and there was a problem with heteroskedasticity in the models. For this reason, standard errors were corrected to be robust to the presence of heteroskedasticity. However, data in logarithmic form were used in the models; thus, the problem of heteroskedasticity was further reduced [79]. Table 3 shows the corresponding

values of $\chi^2$ statistics, the *p*-values for the portmanteau test for correlation in short panels, and the Breusch and Pagan Lagrangian multiplier test.

**Table 3.** Serial correlation and heteroskedasticity tests.

|  | **Model 1** | **Model 2** | **Model 3** |
|---|---|---|---|
| Serial correlation | 35.41 (0.063) | 35.02 (0.066) | 35.55 (0.069) |
| Heteroskedasticity | 120.14 * (<0.001) | 25.83 * (<0.001) | 121.15 * (<0.001) |

* Statistically significant results ($p < 0.05$). Source: Authors' calculation.

Lastly, considering that the number of observation units was 36 and the number of years was 5, according to the central limit theorem, the error term was approximately normally distributed. The estimates of model 3 are presented in the discussion as the basis for the research contribution and conclusions.

## 5. Discussion

Scholars agree that funding opportunities determine the innovation success of SMEs by increasing the level of innovation activities, output performance, productivity, and competitiveness [28,46,52,57]. The novelty of this research lies in its quantitative evaluation of innovation policy programs using a methodology that examines the direct relationship between public funding and an increase in the operating revenues of the beneficiary SMEs.

According to some theoretical contributions, enterprises that received subsidies grew their financial indicators more rapidly [80]. However, there is no evidence that subsidies increased the growth of indebtedness by a certain percentage, according to the findings of this paper. Significant immediate effects on employment and sales were found by the author of [81], whereas positive effects on productivity were only detected 3 years after the treatment group first joined the program. According to certain findings, participation in research projects may increase labor productivity by at least 44.4% while having very little impact on profit margin [82]. There is strong evidence that the public R&D subsidy has a positive impact on the value-added productivity of manufacturing SMEs [83]. The authors in [84] proved that subsidized firms could generate more internal financing and attract more long-term borrowing after receiving the subsidy, but they did not reveal any evidence that subsidized firms could draw more external equity financing than comparable unsubsidized firms.

The research results confirmed that the funds allocated through the Mini-Grants Program and Matching Grants Program positively affect the growth of operating revenues of their beneficiaries. Accordingly, the programs can be considered efficient and useful for enterprises in both early and later stages of development that require funding to accelerate their growth and development. In addition, a 10% increase in corporate indebtedness on average leads to a rise in operating revenue by 6.7%.

Since the total liabilities of enterprises represent indebtedness, their increase is reflected by financial liabilities (commercial bank loans) and operating liabilities. An increase in debt stemming from commercial bank loans can be justified. First, enterprises that use co-financing have a better position with banks. Banks consider these enterprises less risky for the placement of their funds and are more willing to approve their loan applications than they do in the case of enterprises with no co-financing [55]. In addition, enterprises that co-finance part of their business activity through the programs apply for smaller bank loans, which increases their chances. Operating revenue growth is recorded in such enterprises, making access to bank loans even easier [85].

This implies that, for innovative enterprises that used public funding, an increase in indebtedness did not negatively affect profitability, i.e., the tendency to use funds from borrowed funding sources had a positive effect on operating revenue. This can be explained by the fact that, in conditions of limited availability of own funds, which is inherent to

SMEs, especially young enterprises, a more developed mechanism for access to external sources of financing allows the development of innovation capacities [86].

An increase in indebtedness through operating liabilities, primarily liabilities to employees and suppliers, is justified because co-financed business activities attract human capital and new business ventures [87]. On the other hand, attracting quality human capital opens opportunities for developing innovative ideas, innovations, and new business arrangements [88], which shifts part of the financing from bank loans to business liabilities, i.e., liabilities to suppliers.

The obtained result is supported by the existing research, which has shown that directly subsidizing enterprises in their early development stages contributes to surpassing their financial limitations in easier acquisition of additional sources of financing through self-financing or loans from commercial banks [89]. By observing indebtedness, the research concludes that financing restrictions are mitigated through better access to loans from commercial banks and increased business and non-financial liabilities. In addition, operating revenue growth increases the possibility of generating part of the profit in the form of accumulated capital and thus provides conditions for self-financing [90].

Co-financing of enterprises in the initial and early stages of development affects revenue growth and, consequently, performance growth [91], which, while reducing restrictions on obtaining the necessary sources of funding, meets the preconditions for further growth and development of innovation activities, as well as innovation of the beneficiaries of national policy programs. Innovation and innovation activities are among the preconditions for the sustainable development of enterprises [92].

## 6. Concluding Remarks

The research findings provide an economic justification for public funding of micro, small, and medium-sized enterprise innovation activities at both the early and the later phases of development. Specifically, at the micro level, financial support increases business revenues compared to the period before participation in the program. Additionally, the direct link between indebtedness and revenue growth of beneficiaries confirms the positive impact of financial support on their development prospects through facilitating access to financing from both own and borrowed sources. Given that the sustainability of enterprises can be interpreted as achieving development, mitigating financial constraints allows the development of innovation capacities, thus fostering their development and sustainability. At the macro level, achieving sustainable development goals needs competitive economies with a perspective for long-term growth based on innovation.

The research contributes to expanding scientific knowledge about the efficiency of innovation policy programs. To the best of our knowledge, no research on the efficiency of the national innovation programs has been conducted in the Republic of Serbia. Generally, empirical research on the effects of innovation policy programs on the business performance of beneficiary enterprises is scarce. These findings can have important policy implications as they provide guidelines for designing future actions and empirically confirm the need to increase public expenditures for innovation policy. This is particularly relevant in developing countries where innovation policy is not high enough on the agenda. The proposed methodology can be used to evaluate the efficiency of innovation policy measures in Central and Eastern European countries, comparable to Serbia in terms of the availability of innovation funding.

The findings of this research could serve as a theoretical framework for the government's strategy for sustainable innovation subsidies. They provide helpful information for boosting beneficiaries' performance and gathering data for policy makers. Assessing the efficiency of national innovation policy programs may assist policy makers in improving policy governance and implementation. This is important for developing countries whose innovation policy programs are still in the early stages of development.

However, this research evaluated the efficiency of the two programs of Serbia's Innovation Fund with a relatively small number of beneficiaries, which was the study's main

limitation. Future research should be directed at a more comprehensive analysis of the Serbian innovation policy, including more programs and beneficiaries to increase the ability to generalize from the obtained results. A greater sample size would allow for sectoral analysis and concretization of results, i.e., deriving conclusions with broader implications and generating policy suggestions.

A review of the innovation policy of other countries may also be included in future research. This mostly concerns countries that finance innovation through substantially similar programs to Serbia's. When choosing a country for research, it is important to take into consideration additional important aspects that influence the process of implementing innovation policy programs, such as the level of technological development and established relationships, as well as the degree and type of cooperation between all participants in the national innovation system. Such research would considerably improve the reliability of the findings, more effectively assist creators of other countries' innovation policies, and raise new research questions.

**Author Contributions:** Conceptualization, S.Đ. and I.B.; methodology, S.Đ., O.J. and M.P.; validation, S.Đ. and I.B.; formal analysis, S.Đ., I.B., O.J., M.L.-M., M.M.R. and M.P.; investigation, S.Đ. and I.B.; data curation, S.Đ. and O.J.; writing—original draft preparation, S.Đ., I.B., O.J., M.L.-M., M.M.R. and M.P.; writing—review and editing, S.Đ., I.B., O.J., M.M.R., M.L.-M. and M.P.; project administration, S.Đ. All authors have read and agreed to the published version of the manuscript.

**Funding:** The research presented in this paper was funded by the Ministry of Education, Science, and Technological Development of the Republic of Serbia: 451-03-68/2022-14/200005.

**Institutional Review Board Statement:** Not applicable.

**Informed Consent Statement:** Not applicable.

**Data Availability Statement:** Not applicable.

**Conflicts of Interest:** The authors declare no conflict of interest.

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
