# Peer review of "The Efficiency of National Innovation Policy Programs: The Case of Serbia"

_sustainability, doi:10.3390/su14148483_

Round 1

Reviewer 1 Report

This is an interesting paper evaluating EU funding in Serbia. Despite its merits it needs some changes towards its improvement.

1) There are various studies evaluating EU funding and I think that the paper should be enriched with some more resources.

2) It would be good at the first part of the paper to stress (in a couple of lines) the novelty of the paper and what makes unique. 

3) The discussion should be compare the findings with other similar studies. Is the paper verifies our existing knowledge so far or the is a something different?

4) At the concluding part, I think that future research avenues should not be limited in Serbia only but how this can be applied in general in the EU.

Author Response

Dear Reviewer, thank you very much for your suggestions and your kind effort to enhance the quality of our paper. We have carefully and devotedly considered and accepted all your recommendations. The additions and corrections we made in accordance with each of your suggestions are described in detail and presented in the Response document.

Author Response

(The authors gave the same response as above.)

Reviewer 3 Report

Journal Name: Sustainability

Manuscript ID: sustainability-1727202

Paper title: The efficiency of National Innovation Policy Programs: The Case of Serbia

The core subject of the study is given as: 

This paper aims to assess the efficiency of the selected national innovation policy programs in the Republic of Serbia through addressing the following research question:

• Do funds allocated through the Mini-Grants Program and Matching Grants Program affect the operating revenue growth of beneficiary micro, small and medium enterprises?

Abstract, title, and references:

Here are my thoughts on the abstract, title, and references:

1- Abstract:

I am not a big fan, but I totally understand using an unstructured abstract which is typically presented in one paragraph only. However, it conveys the solution to the main research argument, it is clear and valid to the reader and major conclusions and policy implications have been stated.

2- Title:

The title is informative, relevant, and conveys the main idea. 

3- References:

The references are relevant and up to date with 65% between 2016 and 2022. However, the authors should have used a free reference manager like Endnote or Mendeley for the in-text citation. 

In general, the full list of references and in-text cations must be revised to make sure that they are indexed in respected databases such as WoS and/or Scopus.

Introduction/Literature review:

The research introduction is missing. What we have here as the introduction is simply the a very short literature review.

Elements that should be covered in the Introduction:

o State the research problem (purpose of the study).

o State the aims of the study. The following is a list of questions:

Is there a problem? Why is does it exist? Why does it need to be solved?

Who will benefit from the study? In what sense will they benefit?

How will it contribute to what is already known?

An important part of the introduction is where you state the proposal objectives. (After addressing the above questions).

o Provide the context and set the stage for the research question and show its necessity and importance.

o Present the rationale of your proposed study.

o Briefly describe the major issues to be addressed.

o Set the boundaries of the research and its focus.

Furthermore, the author needs to include one more paragraph at the end of the introduction section; the outlined paragraph, for example: “The rest of the study is organized as follows. The next section presents the literature on the ………….., followed by the methodology used. Section four reports the main empirical results, and section five of the article draws conclusions and provides elements for consideration ………”

In addition, the literature review is not sufficient, yet, too brief and needs to be rewritten. Nevertheless, the researcher/s need to give much stronger evidence of the originality of their paper by elaborating more on the literature gap related to this point at the end of the literature review section. However, the conceptual framework is not clear. 

Methods/ Results and Discussion:

Concerning the methodology, the model specification is not clear to the reader not reliable. Furthermore, there are some concerns about the steps of the panel analysis which should be reviewed.

Conclusion:

The conclusion and Recommendation section is sufficient. However, the author needs to extend the section by adding the policy implications related to the study results, limitations (if any), and stating some future research.

Author Response

(The authors gave the same response as above.)

Round 2

Reviewer 3 Report

Thank you for your response.

After a second reading of the manuscript, sustainability-1727202, I still have the following concerns:

1- Articles evaluating EU funding are too many, and I think that the novelty of the paper and what makes it unique is doubtful.

2- Concerning the methodology, the model specification is still not clear to the reader nor reliable. Furthermore, there are some concerns about the steps of the panel analysis which should be reviewed.

3- The discussion should compare the findings with other similar studies. 

Author Response

Dear Reviewer, we have carefully considered and accepted all your recommendations. The additions and corrections we made in accordance with each of your suggestions are described in detail and presented in the Response document.
